# Research on Chemical Mechanical Polishing Technology for Zirconium-Based Amorphous Alloys

**DOI:** 10.3390/mi14030584

**Published:** 2023-02-28

**Authors:** Wei Hang, Chao Song, Ziliang Yin, Ye Liu, Qifan Wang, Yinggang Wang, Yi Ma, Qiaoshi Zeng

**Affiliations:** 1College of Mechanical Engineering, Zhejiang University of Technology, Hangzhou 310023, China; 2Center for High Pressure Science and Technology Advanced Research, Shanghai 201203, China; 3School of Materials Science and Engineering, Southeast University, Nanjing 211189, China; 4Key Laboratory for Light-Weight Materials, Nanjing Tech University, Nanjing 210009, China

**Keywords:** amorphous alloy, chemical mechanical polishing, surface roughness, material removal rate, oxidant

## Abstract

Crystallization often occurs in the processing of amorphous alloys, causing the materials lose their excellent properties. The study adopts chemical mechanical polishing of amorphous alloys, presenting the effect of the rotational speed of the polishing turntable, size of abrasive, polishing pressure, and oxidant concentration. The Taguchi method is used to find the best processing parameters, and AFM is used to characterize the machined material surface. At the same time, XPS is used to detect the change of oxide film composition with the addition of oxidant. The results indicate the optimum process parameters: rotational speed of the polishing turntable is 75 r/min, polishing pressure is 28.3 kPa, the size of abrasive is 0.5 μm, and the size of abrasive is a significant factor affecting surface roughness *Sa*. In addition, as the size of abrasive increases, the material removal rate increases while the surface roughness *Sa* increases. At pH 10, with an abrasive particle size of 0.5 μm, as the H_2_O_2_ concentration increases, the *MRR* first rapidly decreases at 0.21 wt.% H_2_O_2_, and then gradually increases, while the *Sa* decreases. Furthermore, with the addition of oxidant, the main composition of the surface oxide film changes from oxide to hydroxide, and the contents of Zr^4+^ and Cu^0^/Cu^1+^ elements increase. The findings can provide a feasible chemical mechanical polishing process for zirconium-based amorphous alloys to obtain a satisfactory polishing effect.

## 1. Introduction

Amorphous alloy has become a hot topic in the research of advanced metal materials in the past three decades due to their unique amorphous atomic structure. Compared with traditional crystalline metals, amorphous alloys have no structural defects. such as grain boundaries, dislocations, and vacancies, show high strength and hardness close to theoretical values, and good corrosion resistance, wear resistance, and excellent soft magnetic properties; these qualities promote its application in aviation, military, electric power, electronics, and biomaterials. [1,2,3,4]. The development of precision machining methods with high efficiency, low damage, and non-crystallization is crucial for the industrial application of amorphous alloys because the deformation excitation of amorphous alloys is different from that of traditional crystalline metals [3,5]. As such, a more precise association needs to be explored through independent experiments. In the aspect of machining, Han et al. [6] studied the influence of cutting depth, feed speed, and spindle speed on the surface quality of Zr_41.25_Ti_13.75_Ni_10_Cu_12.5_Be_22_ amorphous alloy after turning. They found that, compared with cutting depth and feed speed, spindle speed has the greatest influence on the surface quality after machining. Meanwhile, larger cutting depth is conducive to the formation of regular grooves, while high feed speed increases the instability of the machining process and forms irregular grooves. Xiong et al. [7] studied the turning surface quality and oxidation characteristics of Pd_40_Ni_10_Cu_30_P_20_ amorphous alloy under different cutting parameters and found that the temperature of the cutting zone is in the supercooled liquid zone, that improves the machining surface quality. Meanwhile, a significant amount of cutting heat was produced on the sample surface in the machining process, leading to the oxidation of the material surface. Furthermore, Hsieh et al. [8] studied the machining characteristics of Zr_38.5_Ti1_6.5_Cu_15.25_Ni_9.75_Be_20_ amorphous alloy by Electrical Discharge Machining (EDM) and found that the surface roughness of EDM increases with the increase of current and pulse time. At the same time, the recast layer and carbide produced by EDM increased the surface hardness and kept the hardness of the internal material unchanged. Through orthogonal experiments, Huang et al. [9] explored the performance of micro-EDM of zirconium-based amorphous alloy. Compared with traditional EDM, there was only a weak crystallization peak in the XRD pattern of the material after machining due to its lower discharge energy, and the change of the medium in the machining will also change the crystal phase after machining. By comparing the XRD patterns before and after amorphous alloy abrasive water jet machining with traditional processing methods, Wessels et al. [10] thought that an abrasive water jet could be used for bulk amorphous alloy processing. Ye et al. [11] showed that the material removal of amorphous alloy in the abrasive water jet machining process includes two parts, plastic removal and brittleness removal, and that the abrasive impact behavior is diverse, in which rebound is the primary behavior. These results provide a basis for the study of the erosion mechanism of amorphous alloys. Traditional turning and EDM cannot avoid the crystallization of the material surface during the machining process, while abrasive water jet machining avoids the problem of material crystallization. However, the surface quality of the material after machining is poor, so a machining method that causes less damage to the material while maintaining high machining efficiency is needed for the machining of amorphous alloys.

Chemical mechanical polishing (CMP) is used to reduce machining damage and improve machining quality through the dual action of chemical etching and mechanical grinding. As such, it is widely used in various metal processing fields. Yuan et al. [12] established the material removal rate model of cemented carbide CMP. Under common process conditions, the surface roughness *Ra* is as low as 48 nm. Zhang et al. [13] developed an environmentally friendly chemical mechanical polishing fluid for nickel alloy, whose surface roughness *Ra* reached 0.44nm after processing, and its chemical mechanical polishing mechanism was studied using XPS. Deng et al. [14] studied the effect of pH value and H_2_O_2_ concentration on the chemical mechanical polishing properties of titanium alloy. With the increase in pH value, the material removal rate decreases, and the surface roughness increases. At pH = 4, with the increase of H_2_O_2_ concentration, the surface roughness decreases at first and then increases, and the material removal rate increases rapidly at first and then decreases, reaching the peak at 0.05wt.%. A new type of chemical mechanical polishing solution is optimized by Li et al. [15] to process aluminum alloy; the surface roughness *Ra* decreases from 200 nm to 13 nm, and the material removal rate increases from 150 nm/min to 300 nm/min. Adjusting the concentration of H_2_O_2_ and sodium sulfate is helpful to better balance chemical corrosion and mechanical removal in the processing process, providing a new way for efficient planarization processing of aluminum alloy. However, there are few studies on the introduction of chemical mechanical polishing into the machining of amorphous alloys.

In this paper, one of the most widely used amorphous alloy systems, zirconium-based amorphous alloy, is used as the research object. In response to the problem that conventional processing often fails to achieve flat and non-crystallization processing of amorphous alloy surfaces, a chemical mechanical polishing processing method is used to efficiently obtain high-quality non-crystallization amorphous alloy surfaces using both chemical and mechanical effects. Based on the Taguchi method, in which experiments are designed to study the influence of processing parameters on the processing quality while determining the significant influencing factors through ANOVA, a single analysis is carried out to obtain the best combination of machining parameters. At the same time, XPS is used to explore the effect of oxidant addition on the main components of the passive film to achieve the goal of efficient and high-quality processing on the surface of amorphous alloy.

## 2. Experimental

### 2.1. Sample Preparation

Zr_48_Cu_36_Ag_8_Al_8_ alloy ingots are fabricated by arc melting the mixture of Zr, Cu, Ag, and Al elements in a Ti-gettered high-purity argon atmosphere. All the ingots are melted at least 4 times to ensure their compositional homogeneity. The corresponding 5 mm amorphous columns are prepared using copper mold suction casting, as shown in Figure 3. The physical properties of the Zr_48_Cu_36_Ag_8_Al_8_ amorphous alloy are shown in Table 1.

### 2.2. Experimental Method

First, the initial sample is cut into thin slices of 1mm thickness by a diamond cylindrical cutting machine (SYJ-160, Shenyang Kejing Auto-Instrument Co., Ltd., Shenyang, China), and the surface roughness reaches *Sa* at approximately 120 nm using sandpaper (#2000 SiC, Zhejiang Lixie Instrument Equipment Co., Ltd., Zhejiang, China) for pretreatment. Then the sample is placed on a special precision fixture and processed by a polisher (MoPao3S, Laizhou Weiyi Experimental Machinery Manufacture Co., Ltd., Laizhou, China). The processing diagram is shown in Figure 1. The polishing solution used in the experiment is 5% Al_2_O_3_ (Zhejiang Lixie Instrument Equipment Co., Ltd., Zhejiang, China), the oxidant is H_2_O_2_ (Fuzhou Feijing Biotechnology Co., Ltd., Fuzhou, China), the polishing time is 30 min, the polishing pad is velvet polishing cloth, and the flow rate is precisely controlled by flow pump to 5 mL/min. Since the acidic polishing solution may have the problems of corroding the equipment and polluting the environment, the pH value of the polishing solution is adjusted to 10 by NaOH (Guangzhou Hewei Pharmaceutical Technology Co., Ltd., Guangzhou, China), and the effects of the rotational speed of the polishing turntable, pressure, size of abrasive on the material removal rate, and polishing quality of chemical mechanical polishing of amorphous alloys under alkaline environment are mainly studied.

After the polishing experiment, the samples are ultrasonically cleaned with anhydrous ethanol and deionized water for 10~15 min, then dried with compressed gas for subsequent surface quality and material removal quality testing. The amorphous structure of the sample is determined before and after polishing using an X-ray diffractometer (XRD, Malvern Panalytical Empyrean, Shanghai, China) with a scan range of 10°~90°. Then, the surface roughness information of the polishing sample is obtained using an atomic force microscope (MFP-3D Origin^+^, OXFORD, Changsha, China) and a scanning electron microscope (Versa 3D, FEI) to characterize the surface quality of the machining material. Finally, the X-ray photoelectron spectrometer (AXIS ULTRA^DLD^, Shimadzu, Hongkong, China) is used to analyze the polished amorphous alloy surface and the data are calibrated using the C1s peak (the binding energy is 284.8 eV) for binding energy. Polishing experiments are performed using a precision balance (XSE105DU, Mettler Toledo, Shanghai, China) with the precision of 0.01mg to measure the mass of amorphous alloy before and after polishing. The mass before and after processing are repeated three times to take the average value. The formula for calculating material removal rate (*MRR*) is as follows:(1)$$MRR=\frac{\Delta m}{\rho t\pi d^{2}}$$
where Δ*m* is the mass difference before and after the sample; *ρ* is the density of Zr_48_Cu_36_Ag_8_Al_8_ sample, which is 7.18 g/cm^3^; *d* is the radius of the sample, which is 2.5 mm; and *t* is the polishing time, which is 30 min.

### 2.3. Experimental Design

In the process of chemical mechanical polishing, many factors affect the surface roughness and material removal rate of the processing. According to the analysis of the previous polishing experiments, polishing pressure, rotational speed of the polishing turntable, and size of abrasive are the three primary parameters affecting the polishing effect. Based on the Taguchi method [18] and variance analysis, the processing parameters of amorphous alloy are analyzed and optimized, and the optimum parameters of chemical mechanical polishing of Zr_48_Cu_36_Ag_8_Al_8_ amorphous alloy are determined [19,20,21]. As shown in Table 2, a three-factor and three-level *L*_9_ (3^3^) orthogonal experiment is designed, and then the experimentally obtained signal-to-noise ratio (*S*/*N*) is derived by using the formula. A larger *S*/*N* ratio indicates better quality characteristics, so the parameter with the largest *S*/*N* ratio in the obtained results is the optimal process parameter. In the analysis of *S*/*N*, it is often necessary to select the corresponding quality characteristic function according to the specific index. The quality characteristics are usually divided into three kinds: larger is better, smaller is better, and nominal is better [22]. In this paper, the material removal rate and surface roughness are taken as the index. The material removal rate is larger is better, and the *S*/*N* value is calculated by the Formula (2); the surface roughness is smaller is better, and the *S*/*N* value is calculated by the Formula (3).
(2)$$\eta =-10\log\left[\frac{1}{n}\sum_{i=1}^{n}\frac{1}{y_{i}^{2}}\right]$$
(3)$$\eta =-10\log\left[\frac{1}{n}\sum_{i=1}^{n}y_{i}^{2}\right]$$
where *n* represents the number of experiments in the experimental design, and *y_i_* represents the experimental results obtained under different conditions.

## 3. Results and Discussion

### 3.1. Optimal Factor Level Combination

The *Sa*, *MRR,* and their corresponding *S*/*N* obtained by chemical mechanical polishing of amorphous alloy are shown in Table 3, and the *S*/*N* response plots obtained are shown in Figure 2, that shows the maximum value of *S*/*N* is the optimum level for this factor. From the response plots in Figure 2, it can be seen that with surface roughness *Sa* as the index, the best combination of parameters is A_2_B_3_C_1_ (the rotational speed of the polishing turntable is 75 r/min, pressure is 28.3 kPa, and size of abrasive is 0.5 μm); as such, the average *Sa* obtained is (3.34 ± 0.28) nm, the *MRR* is (405.10 ± 7.09) nm/min. Taking the material removal rate as the index, the best combination is A_3_B_3_C_3_. According to the Preston equation [23], the material removal rate is positively correlated with instantaneous relative velocity *V* and polishing pressure *P*. When the polishing pressure and rotational speed increase, the *MRR* also increases, and although the oversized abrasive particle size increases the material removal rate of the polishing process, it is not conducive to the formation of a smoother surface. As such, the *MRR* is best placed to be sacrificed in order to obtain a flat and smooth surface, and the best combination of parameters is finally experimentally derived as A_2_B_3_C_1_.

Figure 3 shows the XRD pattern of the sample before and after polishing. The sample shows a wide dispersion peak in the range of 30°~45°, and there is no sharp Bragg crystallization diffraction peak, indicating that chemical mechanical polishing can meet the requirements of amorphous alloy non-crystallization processing. Figure 4 shows the AFM images of the polishing experiments. In the range of 10 μm × 10 μm, scratches are produced on the surface of the sample by the scraping of abrasive grains during the polishing process. Due to the different processing parameters, the machined surface quality is also different. Seen in the line outline of the center line of the sample, the surface of the No. 6 experimental group has less undulation and better consistency.

### 3.2. Analysis of Variance

In Taguchi’s experimental data analysis, analysis of variance (ANOVA) can show whether the effect of each control variable on the experimental results is significant [24]. Table 4 and Table 5 show the results of variance analysis for the chemical mechanical polishing experiments, including degrees of freedom, the sum of square deviation of various factors (S.S.), mean square (M.S.), and the ratio of variance F for each factor. This experiment has three factors and three levels, so six degrees of freedom are used to evaluate the influence of different factors, while two degrees of freedom are used as errors. F_0.05_(2,2) = 19, therefore, when the F value in the table is greater than 19, the factor is significant. It can be seen in Table 4 that the rotational speed of the polishing turntable, pressure, and size of abrasive all affect the surface roughness *Sa*, and the degree of influence is size of abrasive > pressure > the rotational speed of the polishing turntable, where size of abrasive is a significant influence, and pressure and the rotational speed of the polishing turntable are non-significant influencing factors [25]. In terms of material removal rate, it can be seen in Table 5 that the degree of influence of each factor is the rotational speed of the polishing turntable > size of abrasive > pressure, while all factors have non-significant influence on the material removal rate of polished samples.

To verify the validity of the experimental results, the verification tests are conducted on the optimal process parameters, and the corresponding experimental results are provided in Table 6. The experimental results show that the differences between the surface roughness results of the validation test are small, so the best surface quality after polishing is achieved when the rotational speed of the polishing turntable is 75 r/min, the pressure is 28.3 kPa, and size of abrasive is 0.5 μm.

### 3.3. Effect of Size of Abrasive on Sa and MRR

In the chemical mechanical polishing of metals, the material removal rate of the workpiece surface increases with the increase of abrasive grain size [26]. The larger the abrasive grain size, the higher the material removal rate during polishing, however the final processing surface is worse, due to the larger grain size of abrasives in producing deeper scratches. On the contrary, the smaller grain size of abrasives causes the material removal rate to be lower, but with a better surface quality. Surface roughness and material removal rate are the two major indicators that must be considered during polishing. It can be seen from the ANOVA that the size of abrasive has a significant effect on the surface roughness of the polished materials. To further study the effect of size of abrasive on surface roughness and material removal rate, polishing experiments are conducted with three different sizes of Al_2_O_3_ abrasives, and the polishing results are shown in Figure 5. When the abrasive size decreases from 1.5 μm to 0.5 μm, the material removal rate decreases from 812.57 ± 3.05 nm/min to 405.10 ± 7.09 nm/min, a reduction of 50.15%, and the surface roughness decreases from 4.42 ± 0.61 nm to 3.34 ± 0.28 nm, a reduction of 24.43%. This is primarily because the smaller the abrasive size, the smaller the processing area when other processing parameters are consistent [27]. At the same time, the smaller the amount of material removed during processing, the less the machined surface is affected by deeper scratches affecting its surface quality. Figure 6 shows the AFM diagram of the polishing sample surface. From its line outline, it can be seen that the larger the abrasive size, the greater the undulation of its profile and the worse the surface quality. Therefore, among the three different abrasives with different sizes selected for the experiment, the Al_2_O_3_ abrasive with size of 0.5 μm has the best processing effect.

### 3.4. Effect of H_2_O_2_ Concentration on Sa and MRR

Li et al. [28] found that, during the process of chemical mechanical polishing of metal materials, the oxidant can form an oxide film on the metal surface. This oxide film is removed under the action of abrasives, and the process is repeated to achieve material removal. Under the condition of pH = 10, the polishing solution with H_2_O_2_ concentrations of 0 wt.%, 0.09 wt.%, 0.21 wt.%, and 0.3 wt.% are configured to study the effect of oxidant concentration on the material removal rate and surface roughness *Sa* in the polishing process; the experimental results are shown in Figure 7. With the increase of oxidant concentration, the material removal rate decreases rapidly at first and then increases. When the H_2_O_2_ concentration is 0.21 wt.%, the minimum value is 274.27 ± 6.10 nm/min. This is due to the fact that, with the addition of oxidant, a passivation film is generated on the surface of the material, resulting in a rapid decrease in the material removal rate. Then, with the further increase in the concentration of oxidant, the passivation film becomes dense [13] while its hardness decreases. When the oxidant concentration is 0.21 wt.%, the balance between film formation and mechanical removal is reached, and the material removal rate is the lowest. With further increases in the oxidant, the mechanical removal effect is higher than the film formation and the material removal rate increases to 405.10 ± 7.09 nm/min. Surface roughness decreases gradually with the increases of oxidant concentration, from 4.77 ± 0.28 nm at 0 wt.% H_2_O_2_ concentration to 3.34 ± 0.28 nm at 0.3 wt.%, a reduction of 29.98%. As can be seen in Figure 8, the best surface is obtained when the H_2_O_2_ concentration is 0.3 wt.%.

Figure 9 shows the SEM image of the sample surface before and after polishing. It should be noted that, since the initial sample is obtained by copper mold suction casting, the surface of the sample has defects, such as bumps, pits, and pores, that greatly increase the surface roughness of the sample. After chemical mechanical polishing, the surface of the sample is flat and smooth without obvious processing defects, which verified the feasibility of obtaining a high-quality amorphous alloy surface by chemical mechanical polishing.

### 3.5. XPS Analysis

To explore the role of H_2_O_2_ oxidant in the passivation film formation process, the polishing samples with 0 wt.% and 0.3 wt.% oxidant concentration are subjected to XPS analysis. Since the selected materials accounted for the largest proportion of Zr and Cu, the analysis focused on the valence changes of Zr, Cu, and O elements in the surface oxide film, whose elemental fine spectra are shown in Figure 10.

As shown in Figure 10a, the O 1s spectrum consists of two peaks with peak positions of 530 eV and 531.6 eV, where the binding energy of 530 eV belongs to O^2−^, while the binding energy of 531.6 eV comes from OH^−^ [29,30]. With the addition of oxidant, the intensity of the OH^−^ peak is significantly enhanced, and the passivation film composition gradually changed from the original oxide dominated to hydroxide dominated. The Zr 3d spectrum consists of four peaks, as shown in Figure 10b, with Zr^0^ peaks at 179 eV and 181.4 eV and Zr^4+^ peaks at 182.2 eV and 184.6 eV [30,31]. The addition of the oxidant increases the ratio of Zr^0^ to Zr^4+^ on the sample surface from 1:33.28 to 1:38.1. The elemental Cu changes similarly, with Cu 2p spectrum consisting of two peaks, as shown in Figure 10c. The peak positions on this spectrum are 932.4 eV and 933.3 eV without the addition of oxidant, and 932.7 eV and 933.4 eV with the addition of oxidant. This corresponds to Cu^0^/Cu^1+^ and Cu^2+^, respectively [32]. With the addition of oxidant, the ratio of Cu^0^/Cu^1+^ to Cu^2+^ content on the sample surface changed from 1.05:1 to 1.3:1. All these changes originated from the strongly oxidizing properties of the oxidant, that increased the Zr^4+^ and Cu^0^/Cu^1+^ elements in the surface oxide film of the sample. The corresponding reaction equations are shown in Equations (4)–(7).
(4)$$Zr+2H_{2}O_{2}\to ZrO_{2}+2H_{2}O$$
(5)$$Zr+4H_{2}O_{2}\to Zr{\left(OH\right)}_{4}\to Zr^{4+}+4OH^{-}$$
(6)$$2Cu+2H_{2}O_{2}\to Cu_{2}O+H_{2}O$$
(7)$$Cu_{2}O+2H_{2}O_{2}\to 2CuO+H_{2}O$$

Based on the results of polishing and characterization, the material removal mechanism of zirconium-based amorphous alloy in the CMP is proposed, as shown in Figure 11. In the alkaline environment with pH 10, the oxidation reaction continuously produces Zr^4+^ and Cu^2+^, resulting in a passive film on the surface of the sample. This film inhibits the further reaction of the material, and the material is removed by abrasive mechanical removal. With the addition of oxidant, the oxidation reaction increases, the content of hydroxide in the passive film increases, and the wear resistance of the material increases. Therefore, compared with the sample without oxidant, the material removal rate and surface roughness are decreased, and the surface scratches are reduced. As a result, a smooth surface can be obtained.

## 4. Conclusions

Chemical mechanical polishing was applied to the polishing of zirconium-based amorphous alloys. The effects of the rotational speed of the polishing turntable, pressure and size of abrasive on polishing quality, and material removal rate under alkaline conditions were investigated through experimental design and ANOVA using the Taguchi method. The optimal combination of process parameters was obtained, while single-factor experiments were conducted on the significant influencing factors affecting the polishing quality. The following conclusions were obtained.
(1)The orthogonal experiments show that the polishing surface roughness *Sa* is minimized when the rotational speed of the polishing turntable is 75 r/min, the polishing pressure is 28.3 kPa, and the size of the abrasive is 0.5 μm. XRD patterns of samples before and after polishing are amorphous structures, indicating that chemical mechanical polishing can meet the requirements of efficient non-crystallization processing of amorphous alloys.(2)The material removal rate and surface roughness decreased with the reduction of particle size. The material removal rate decreased from 812.57 ± 3.05 nm/min to 405.10 ± 7.09 nm/min, a reduction of 50.15%. The surface roughness decreased by 24.43% from 4.42 ± 0.61 nm to 3.34 ± 0.28 nm. With the increase of H_2_O_2_ concentration, the material removal rate decreased rapidly and then increased. When the concentration was 0.21 wt.%, the minimum value was 274.27 ± 6.10 nm/min, while the surface roughness decreased with the increase of concentration, reaching the minimum value of 3.34 ± 0.28 nm at 0.3 wt.%.(3)XPS analysis shows that the oxide film on the sample surface is composed of oxides and hydroxides. With the addition of oxidants, the oxidation and wear resistance of the samples are enhanced, and the main components are transformed into hydroxides. At the same time, the contents of Zr^4+^ and Cu^0^/Cu^1+^ also increase. The results can provide some reference for chemical mechanical polishing zirconium-based amorphous alloys.


## Figures and Tables

**Figure 1 micromachines-14-00584-f001:**
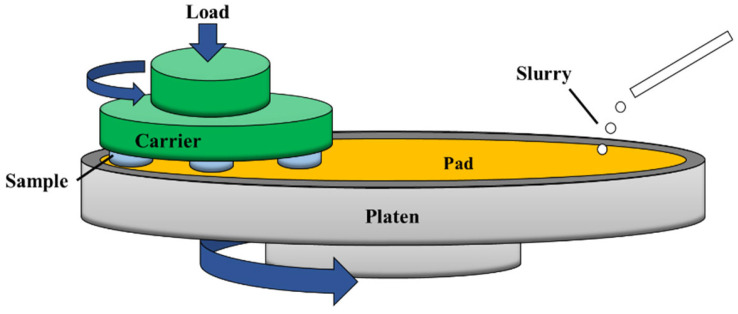
Polishing schematic diagram.

**Figure 2 micromachines-14-00584-f002:**
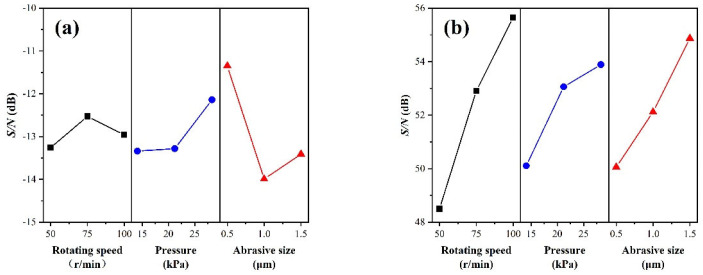
Plots of the control factor effects. (**a**) *Sa*; (**b**) *MRR*.

**Figure 3 micromachines-14-00584-f003:**
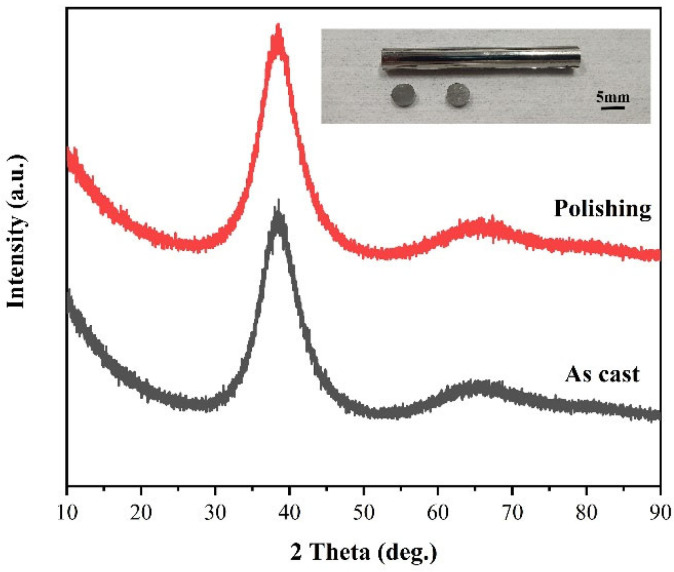
X-ray diffraction patterns of the Zr_48_Cu_36_Ag_8_Al_8_ sample before (black) and after (red) polishing.

**Figure 4 micromachines-14-00584-f004:**
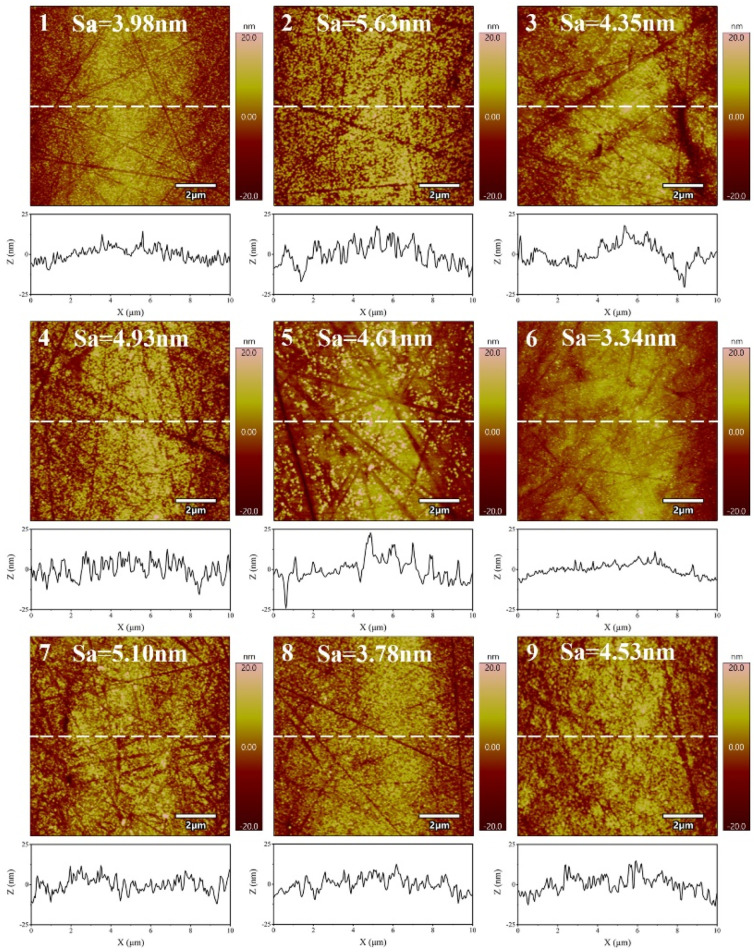
AFM images of amorphous alloy surfaces after the polishing experiment.

**Figure 5 micromachines-14-00584-f005:**
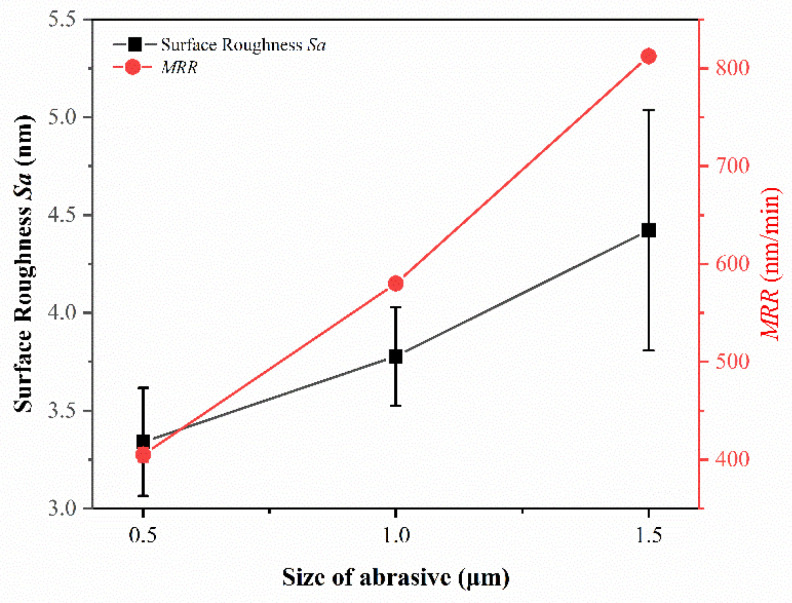
Effect of size of abrasive on the surface roughness and *MRR* of polished samples.

**Figure 6 micromachines-14-00584-f006:**
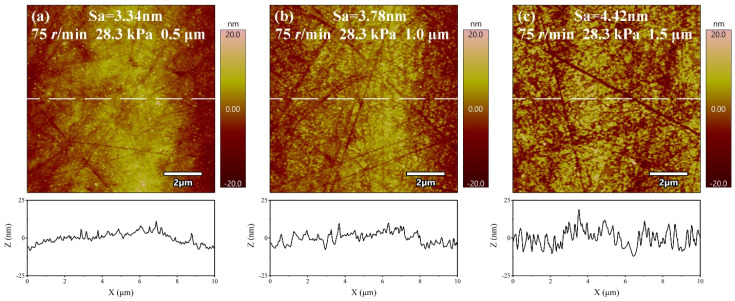
AFM images of amorphous alloy surfaces after polishing experiments with different particle sizes. (**a**) 0.5 μm; (**b**) 1 μm; (**c**) 1.5 μm.

**Figure 7 micromachines-14-00584-f007:**
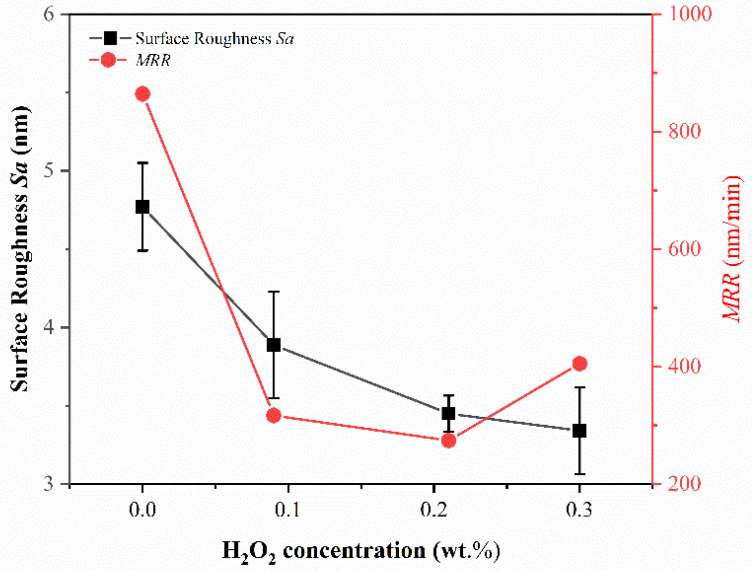
Effect of H_2_O_2_ concentration on surface roughness and material removal rate of polished samples.

**Figure 8 micromachines-14-00584-f008:**
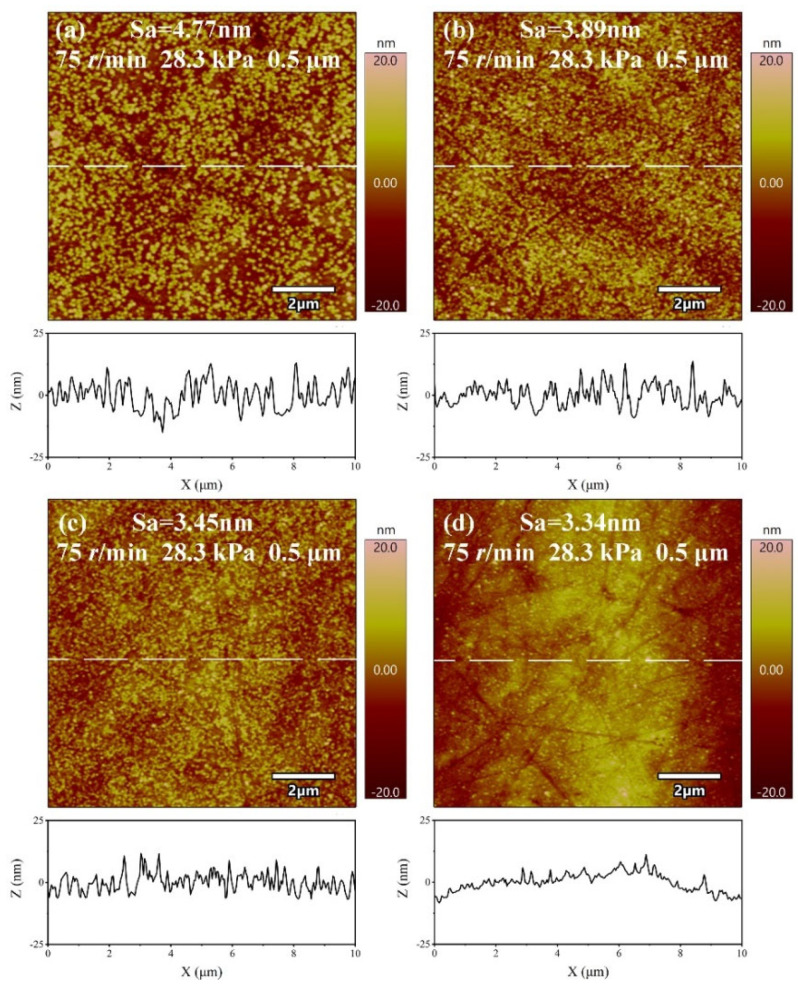
AFM images of amorphous alloy surfaces after polishing experiments with different H_2_O_2_ concentrations. (**a**) 0 wt.%; (**b**) 0.09 wt.%; (**c**) 0.21 wt.%; (**d**) 0.3 wt.%.

**Figure 9 micromachines-14-00584-f009:**
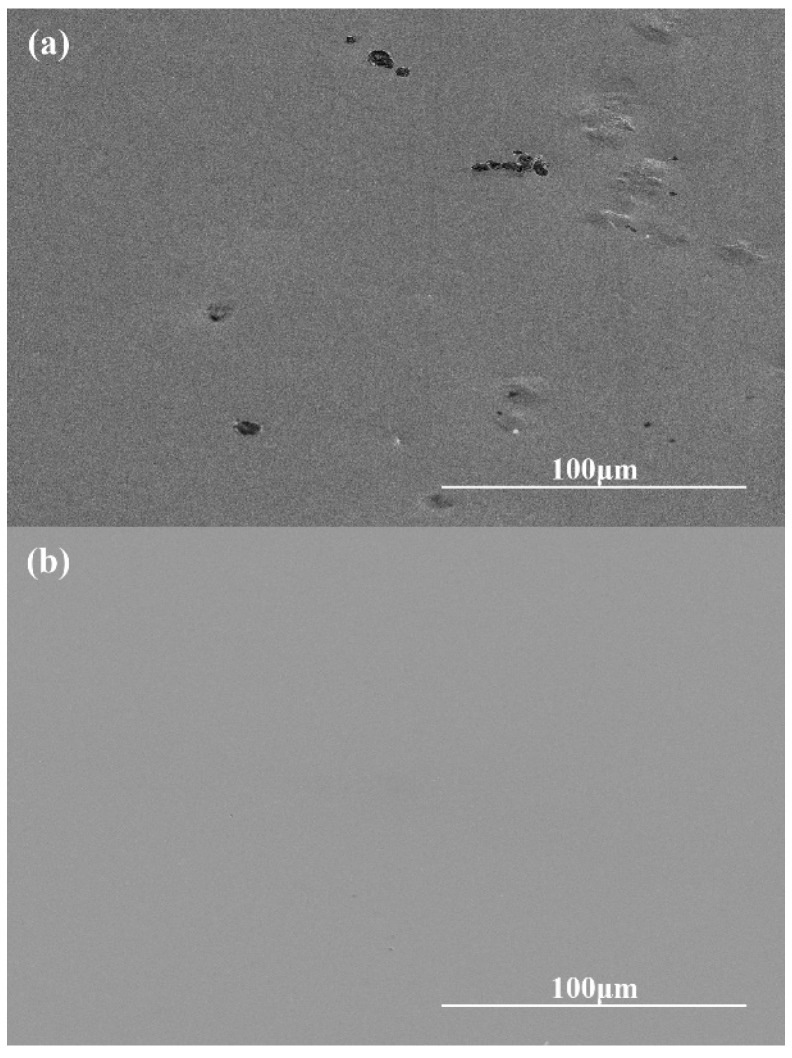
SEM images of the sample surface. (**a**) before polishing; (**b**) after polishing.

**Figure 10 micromachines-14-00584-f010:**
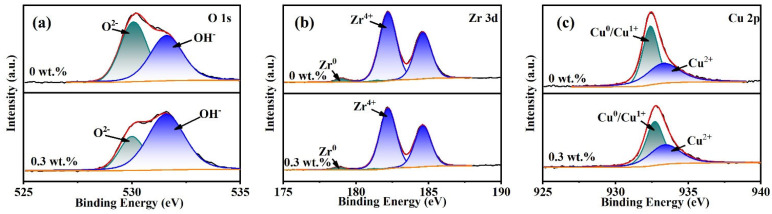
XPS narrow spectra of polished sample surfaces with H_2_O_2_ concentration of 0 wt.% and 0.3 wt.%. (**a**) O1s; (**b**) Zr 3d; (**c**) Cu 2p.

**Figure 11 micromachines-14-00584-f011:**
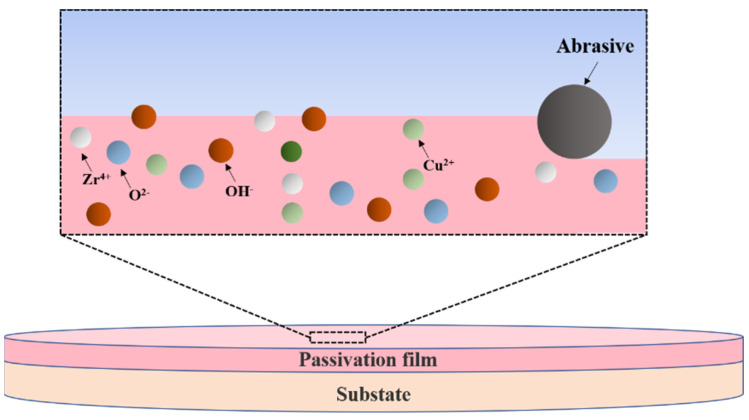
Schematic illustration of the material removal principle.

**Table 1 micromachines-14-00584-t001:** Properties of the Zr_48_Cu_36_Ag_8_Al_8_ amorphous alloys.

Parameters	Values
Density (g/cm^3^)	7.18
Hardness (MPa)	7.20
Yield strength (MPa)	1880 [16]
Fracture strength (MPa)	1903 [16]
*T*_g_ (K, Glass transition temperature)	683 [17]
*T*_x_ (K, Crystallization temperature)	791 [17]

**Table 2 micromachines-14-00584-t002:** The control factors and their levels in polishing experiments.

Level	A. Rotating Speed (r/min)	B. Pressure (kPa)	C. Abrasive Size (μm)
1	50	14.1	0.5
2	75	21.2	1.0
3	100	28.3	1.5

**Table 3 micromachines-14-00584-t003:** Experimental results of the surface roughness, material removal rate and *S*/*N* values.

Experiment. No.	Control Factor			Surface Roughness *Sa* (nm)				*MRR* (nm/min)	*S*/*N* Ratio(dB)	
	A	B	C	1	2	3	Mean (nm)		*Sa*	*MRR*
1	1	1	1	3.93	3.78	4.22	3.98	157.63	−12.00	43.95
2	1	2	2	5.64	5.57	5.67	5.63	280.58	−15.00	48.96
3	1	3	3	4.20	4.57	4.29	4.35	425.20	−12.78	52.57
4	2	1	2	4.87	5.20	4.68	4.93	333.38	−13.84	50.46
5	2	2	3	4.55	4.73	4.55	4.61	640.76	−13.27	56.13
6	2	3	1	3.45	3.15	3.42	3.34	405.10	−10.49	52.15
7	3	1	3	5.33	4.50	5.47	5.10	624.21	−14.18	55.91
8	3	2	1	3.88	3.89	3.58	3.78	505.99	−11.56	54.08
9	3	3	2	4.80	4.22	4.58	4.53	703.81	−13.14	56.95

**Table 4 micromachines-14-00584-t004:** ANOVA of the surface roughness.

Factor	D.F.	S.S.	M.S.	F Value	F_0.05_(2,2)
A	2	0.80	0.40	1.87	19
B	2	2.76	1.38	6.43	19
C	2	11.61	5.81	27.04	19
Error	2	0.43	0.21	-	-
Total	8	15.61	-	-	-

**Table 5 micromachines-14-00584-t005:** ANOVA of the material removal rate.

Factor	D.F.	S.S.	M.S.	F Value	F_0.05_(2,2)
A	2	157,216	78,608	14.82	19
B	2	31,587	15,794	2.98	19
C	2	65,211	32,606	6.15	19
Error	2	10,611	5306	-	-
Total	8	264,626	-	-	-

**Table 6 micromachines-14-00584-t006:** Sample polishing results of the confirmation trials.

Experiment. No.	Surface Roughness *Sa* (nm)				*S*/*N* Ratio (dB)
	1	2	3	Mean (nm)	
1	3.71	3.26	3.58	3.42	−10.94
2	3.96	3.13	3.22	3.44	−10.78
3	3.49	3.17	3.22	3.29	−10.35

## Data Availability

Data are only available upon request due to restrictions regarding.

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
