# Peer review of "Research on Chemical Mechanical Polishing Technology for Zirconium-Based Amorphous Alloys"

_micromachines, 2023, doi:10.3390/mi14030584_

Round 1

Reviewer 1 Report

In this paper, the authors investigate chemical mechanical polishing for machining zirconium-based amorphous alloys, perform orthogonal tests to optimize the machining process, and use XPS to discuss and analyze the material removal mechanism during the machining process, which is significant for amorphous alloy machining. The whole paper is well organized, and the research work amount is impressive. The results and conclusions of the paper are valuable. I would advise that the paper can be accepted after minor revision. The following discussions and suggestions can be considered by the authors:

1.     The importance of the studied issue should be further highlighted.

2.     Why choose aluminum oxide used for abrasive?

3.     Why abrasive particle size, pressure and rotational speed are selected for orthogonal test?

4.     In the article, the full name of the noun should appear before the abbreviation.

5.     More detailed information should be shown in Fig. 6 and 8.

6.     The description of fig. 8 should be added appropriately.

7.     Why does the hardness of the surface oxide film decrease with increasing oxidant concentration?

8.     Fig. 10 is a little blurry. It should be considered to replacing it with clearer one.

9.     The ratio of abrasive to hydroxide should be noted in Fig. 11

10.  The third point of the conclusion, the elemental valence is not superscripted.

Author Response

Point 1: The impotance of the studied issue should be further highlighted.

Response 1:

According to the point 1, the author analyzes and discusses the references of amorphous alloy processing in the introduction, highlights the advantages and significance of chemical mechanical polishing of amorphous alloys, and further emphasizes the importance of the problems studied.

Point 2: Why choose aluminum oxide abrasives?

Response 2:

There are three kinds of abrasives commonly used in chemical mechanical polishing, which are alumina, silicon dioxide and ceria. Compared with the other two common abrasives, alumina abrasives are characterized by high hardness and high material removal rate in the machining process. This meets the requirements of efficient processing in this paper. At the same time, the materials studied in this paper have high hardness and are suitable for processing with alumina abrasives, so the abrasive selected in this experiment is alumina.

Point 3: Why abrasive particle size, pressure and rotational speed are selected for orthogonal test?

Response 3:

According to the material removal rate model, the material removal rate in the polishing process is not only related to the properties of abrasive particles, polishing pads and samples, but also related to polishing speed and polishing pressure. The depth model of the material pressed into the sample shows that it is related to the particle size of the abrasive. In this paper, the material removal rate and surface roughness are taken as the index, so the abrasive particle size, pressure and rotational speed are selected for orthogonal test.

Point 4: In the article, the full name of the noun should appear before the abbreviation.

Response 4:

The author has revised the related problems in the article.

Point 5: More detailed information should be shown in Fig. 6 and 8.

Response 5:

We have redrawn and re-labeled Figure 6 and 8.

Point 6: The description of fig. 8 should be added appropriately.

.

Response 6: An explanation of Figure 8 has been added

Point 7: Why does the hardness of the surface oxide film decrease with increasing oxidant concentration?

Response 7:

It can be found from the literature that the hardness of the oxide film formed during the processing of metal materials decreases with the increase of the concentration of oxidant. The essence of zirconium based amorphous alloy is still a kind of metal material, and it can be seen from the material removal rate trend that the surface hardness decreases with the increase of the concentration of oxidant.

Point 8: Fig. 10 is a little blurry. It should be considered to replacing it with clearer one.

Response 8:

We have redrawn and re-labeled Figure 10.

Point 9: The ratio of abrasive to hydroxide should be noted in Fig. 11

Response 9:

We have redrawn and re-labeled Figure 11.

Point 10: The third point of the conclusion, the elemental valence is not superscripted

Response 10:

We have corrected the problems in the conclusion and reviewed the full text

Reviewer 2 Report

In the manuscript, the authors adopted chemical mechanical polishing of amorphous alloys to investigate the effect of the rotational speed of the polishing turntable, size of abrasive, polishing pressure, and oxidant concentration. The results are interesting and significant for the application of this kind of alloys. The paper can be published after the following concerns were addressed:

1. Why did the authors choose the Zr48Cu36Ag8Al8 for present study? This composition contains the novel metal element Ag, which may not suitable for the massive production and application. 

2. It can be seen from Fig. 1 that several samples were polished together under the same the polishing conditions, what about the uniformity of each sample? Does the authors have evidences? 

3.From Fig. 4, it can be seen that the surface roughness of the samples after polishing was around 5 nm, can this value be even improved to less than 1 nm?

4. During the polishing process, is it possible that the local temperature goes up into its supercooled liquid region?

5. In the introduction section, the related references should be discussed and compared on the machining of metallic glasses, such as Micromachines 202011(1), 4Mater. Futures 1 012001 et al. 

Author Response

Point 1: Why did the authors choose the Zr48Cu36Ag8Al8 for present study? This composition contains the novel metal element Ag, which may not suitable for the massive production and application. 

Response 1:

Thank you for your question. The zirconium-based amorphous alloy system selected in this paper is Zr48Cu36Ag8Al8, and the amorphous alloy with a diameter of 2 mm is selected for the experiment, so there are requirements for the forming ability of the amorphous alloy, and this system has good amorphous forming ability and high thermal stability, so it is easy to prepare bulk amorphous alloy samples for experiments to ensure the consistency of the experimental samples. The element of Ag in the composition is low, and the quality of Ag needed in the preparation process is less. Combined with its excellent properties, the author thinks that Ag will not become an obstacle to its production and application.

Point 2:  It can be seen from Fig. 1 that several samples were polished together under the same the polishing conditions, what about the uniformity of each sample? Does the authors have evidences? 

Response 2:

Thank you for your valuable comments. In this paper, several samples are polished together under the same parameters to ensure the reliability of the experimental data. The experimental data are all the average values after data processing, and the deviations of the polishing results of multiple samples have been shown in the figure in the form of error rods, such as figures 6 and 8. At the same time, this paper also sets up a set of validation tests, and the surface roughness information obtained is shown in Table 6. It can be found that the polishing results are consistent under the same parameters.

Point 3: From Fig. 4, it can be seen that the surface roughness of the samples after polishing was around 5 nm, can this value be even improved to less than 1 nm?

Response 3:

Thank you for your valuable comments. In this paper, the processing parameters of chemical mechanical polishing zirconium-based amorphous alloy are optimized, and the minimum surface roughness is 3.34 nm. In reading the relevant literature, the author found that the use of silicon dioxide polishing solution can further reduce the surface roughness, such as silicon wafer and silicon carbide wafer polishing. Therefore, the author believes that if the silicon dioxide polishing solution is used for secondary processing, the surface quality can be further optimized to below 1 nm.

Point 4: During the polishing process, is it possible that the local temperature goes up into its supercooled liquid region?

Response 4:

Thank you for your valuable comments. In this paper, chemical mechanical polishing is used to process amorphous alloy, the abrasive particle size in the polishing slurry is very small, and the depth of its pressing into the material in the machining process is also very small, so it will not produce too much cutting heat. At the same time, the polishing slurry also plays a cooling role in the machining process, taking away the chip and excess heat, so the temperature will not rise to the supercooled liquid zone.

Point 5: In the introduction section, the related references should be discussed and compared on the machining of metallic glasses, such as Micromachines 202011(1), 4, Mater. Futures 1 012001 et al. .

Response 5:

Thank you for your valuable comments. The author has revised the references in the introduction of this article and discussed and compared them.

Round 2

Reviewer 2 Report

The manuscript should be published now.